# Stress–Strain Behavior of FRC in Uniaxial Tension Based on Mesoscopic Damage Model

**Weifeng Bai, Xiaofeng Lu, Junfeng Guan \*, Shuang Huang \*, Chenyang Yuan and Cundong Xu**

School of Water Conservancy, North China University of Water Resources and Electric Power, Zhengzhou 450046, China; baiweifeng@ncwu.edu.cn (W.B.); z20201010121@stu.ncwu.edu.cn (X.L.); yuanchenyang@ncwu.edu.cn (C.Y.); xucundong@ncwu.edu.cn (C.X.)
\* Correspondence: junfengguan@ncwu.edu.cn (J.G.); z201710103065@stu.ncwu.edu.cn (S.H.)

**Abstract:** Fiber-reinforced concrete (FRC) is widely used in the field of civil engineering. However, the research on the damage mechanism of FRC under uniaxial tension is still insufficient, and most of the constitutive relations are macroscopic phenomenological. The aim is to provide a new method for the investigation of mesoscopic damage mechanism of FRC under uniaxial tension. Based on statistical damage theory, the damage constitutive model for FRC under uniaxial tension is established. Two kinds of mesoscopic damage mechanisms, fracture and yield, are considered, which ultimately determines the macroscopic nonlinear stress–strain behavior of concrete. The yield damage mode reflects the potential bearing capacity of materials and plays a key role in the whole process. Evolutionary factor is introduced to reflect the degree of optimization and adjustment of the stressed skeleton in microstructure. The whole deformation-to-failure is divided into uniform damage phase and local failure phase. It is assumed that the two kinds of damage evolution follow the independent triangular probability distributions, which could be represented by four characteristic parameters. The validity of the proposed model is verified by two sets of test data of steel fiber-reinforced concrete. Through the analysis of the variation law of the above parameters, the influence of fiber content on the initiation and propagation of micro-cracks and the damage evolution of concrete could be evaluated. The relations among physical mechanism, mesoscopic damage mechanism, and macroscopic nonlinear mechanical behavior of FRC are discussed.

**Keywords:** fiber-reinforced concrete; damage mechanism; uniaxial tension



## 1. Introduction

Fiber-reinforced concrete (FRC) has been widely used in the field of civil engineering for its excellent physical and mechanical properties. FRC is a kind of cement-based composite material composed of metal fiber, inorganic non-metallic fiber, synthetic fiber, or natural organic fiber as a reinforcing material. The most widely used are steel fiber-reinforced concrete (SFRC) and polypropylene fiber-reinforced concrete (PFRC). A large number of experimental studies have shown that these randomly distributed fibers can effectively hinder the expansion of micro-cracks in the concrete and the formation of macro-cracks, significantly improving the compressive, tensile, bending, impact resistance, and fatigue resistance of concrete [1–12]. Fiber materials are further used in the research of new types of concrete, such as coral concrete, geopolymer concrete, self-compacting concrete, etc. Liu et al. [7] carried out uniaxial compression tests of carbon fiber-reinforced coral concrete and established empirical piecewise constitutive model. At present, the representative theories on the mechanism of fiber reinforcement mainly include the fiber spacing theory proposed by Romualdi and Batson [13], and the reinforcement rules for composite materials proposed by Swamy [14].

The constitutive relation of concrete material is one of the key problems in the nonlinear analysis of concrete structure. Much effort has been devoted in the last decade to model the FRC by considering it as a composite material composed of concrete matrix

and fibers [15–19]. Uniaxial stress–strain behavior is the most fundamental constitutive relationship of the concrete material. Due to the limitation of experimental technology and the deficiency of relevant theories, the research achievements on uniaxial compression are abundant, while the research on tension is relatively few. The research on the constitutive relation of FRC is based primarily on the relevant theoretical results of ordinary concrete, mainly focusing on the influence of fiber characteristic parameters on the constitutive relation. In recent years, based on the direct tensile test carried out on the universal test machine, the uniaxial tensile stress–strain curves for different types of FRC were obtained, and the corresponding macroscopic phenomenological constitutive models were proposed [3,8,20–22].

As a typical quasi-brittle material, the nonlinear macroscopic stress–strain behavior of concrete are closely related to the heterogeneity in microstructure. The essential nature lies not only in the initiation and propagation of individual micro-cracks, but also in the interaction and coalescence of crack populations [23,24]. Most of the existing concrete constitutive relations are macroscopic phenomenological models established by polynomial and other mathematical formulas, focusing on the fitting of test data, but lacking the clear physical meaning for the parameters. Therefore, it is difficult to reveal the meso-damage evolution law of concrete in essence by those phenomenological models.

Damage Mechanics (DM) [25–27] is a relatively new field studying the response and reliability of materials with countless randomly distributed irregular microcracks. The fundamental aspect of damage mechanics is the selection of damage parameters. The essential feature of the original Kachanov's model [28] resides in the introduction of a special internal variable defined by the state of local damage and its accumulation. Many macroscopic continuum damage models and micromechanics damage models [25,29] were proposed after Kachanov's work. Li et al. [21] suggested a continuum damage mechanics-based model for FRC in tension, in which the quasi-brittleness of the matrix and the fiber–matrix interfacial properties were taken into consideration. Considering the stochastic characteristics of the distribution of disfigurements in the microstructure, a physically motivated damage model (i.e., the bundle parallel bar system (PBS)) was proposed by Krajcinovic and Silva [30], based on the continuous damage theory and statistical strength theory. After this work, Statistical Damage Mechanics (SDM) was gradually formed, constitutive relations of quasi-brittle material (rock and concrete) have been extensively studied, and some inspiring results were obtained [31–34]. This kind of model abstracts quasi-brittle material into a complex system composed of mesoscopic physical elements. The heterogeneity in the microstructure is introduced by assuming that the characteristic parameters of the mesoscopic unit follow Weibull or other statistical distribution forms. The progressive damage accumulation process of concrete is described by means of probability and statistics. It ignores the complicated physical details of the damage process and avoids the complicated calculation of statistical mechanics. The relationship between meso-damage mechanism and macroscopic mechanical behavior of material is effectively established. Considering there are two fundamental damage modes (fracture and yield) in the microstructure of concrete, the statistical damage models of concrete under uniaxial and multiaxial loading were proposed by Chen et al. [35] and Bai et al. [36–38], which could be used to predict the constitutive behavior of concrete under a complex loading environment and explore the damage mechanism of the concrete material.

In this paper, a statistical damage constitutive model for the FRC is presented based on the macroscopic test phenomenon and the statistical damage theory. Firstly, a deep analysis of the mesoscopic damage mechanism for concrete material under uniaxial tension is elaborated. It indicated that the macroscopic nonlinear stress–strain behavior is determined by the evolution of fracture and yield damage on a meso scale. The yield damage mode, reflecting load redistribution and adjustment of stress skeleton in the microstructure, is emphasized as the key role in the whole deformation and failure process. Evolutionary factor is introduced to reflect the development of the potential mechanical capacity of materials. Subsequently, the triangular probability distributions with four parameters are used

to simulate the cumulative evolution of fracture and yield damage. By defining the above parameters as a function of fiber content, the impact of fiber content on the nucleation and growth of microcracks and the mesoscopic damage evolution of the FRC is reflected. The determination method for the model's characteristic parameters is proposed. The validity of the proposed model is demonstrated by comparing the theoretical and experimental results. The influence rule of fiber content on the mesoscopic damage evolution of the FRC is analyzed.

## 2. Deformation and Failure of Concrete under Uniaxial Tension

Since the failure of concrete is essentially caused by the nucleation and growth of microcracks produced by local tensile strain, the uniaxial tension could be regarded as the most fundamental failure form for concrete.

According to the macroscopic experimental phenomena, the deformation and failure of the concrete specimen under uniaxial tension seems to be divided into two stages, the uniform damage stage (US) and the local fracture stage (LS), as shown in Figure 1a. In the uniform damage stage, the whole specimen remains uniformly loaded and deformed, with the damage evolving mainly by the nucleation and growth of micro-defects, which randomly distribute in the whole specimen. In the local fracture stage, a macroscopic main-crack perpendicular to the tensile direction forms in the fracture process zone (FPZ). The macroscopic response of the specimen strongly depends on the size of the largest crack with a preferential orientation in the FPZ. Local further damage and fracture occurs in the FPZ, while the rest of the region of the specimen shows the unloading behavior.

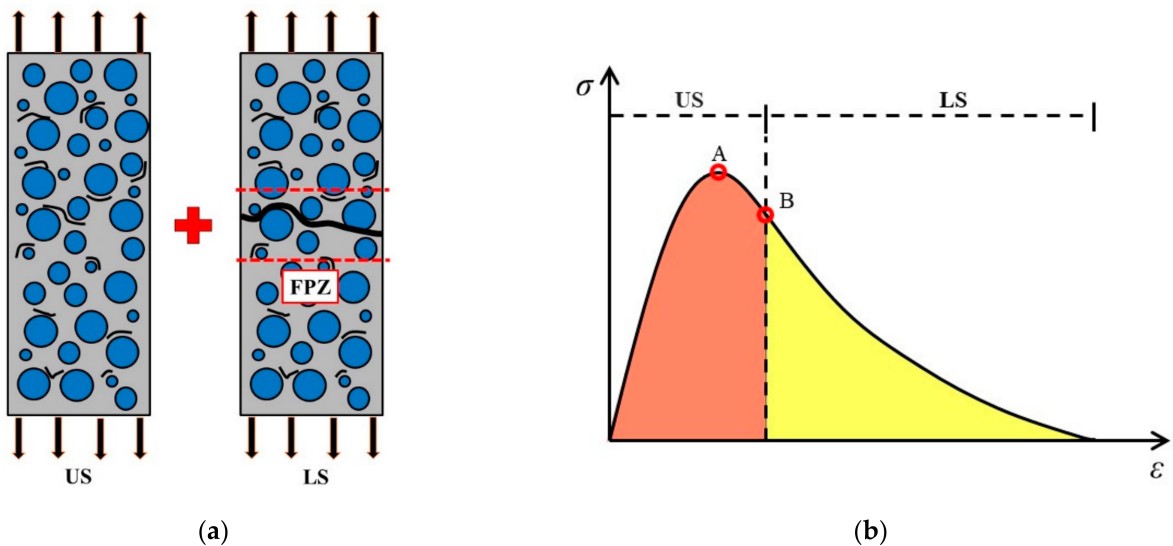

(**a**)            (**b**)

**Figure 1.** Two-stage feature of macroscopic mechanical behavior of concrete under uniaxial tension: (**a**) deformation-failure process; (**b**) typical nominal stress–strain curve.

As shown in Figure 1b, a typical master stress–strain curve of concrete specimen under uniaxial tension is presented. The signs A and B denote, respectively, the two characteristic states, i.e., (1) the peak nominal stress state and (2) the critical state at which macro-crack starts to develop in the FPZ and the damage process switches into the local failure stage. From the shape of the curve, the stress–strain curve seems to be divided into a stress-strengthening phase and a strain-softening phase bounded by state A. Before A, the tensile stress increases gradually with the growth of tensile strain. After A, the tensile stress decreases gradually with the increase of tensile strain, until to zero. The stress value corresponding to state A, known as the so-called tensile strength, is considered to be one of the most important mechanical indexes for concrete. The descending segment of the tensile stress–strain curve for concrete was firstly measured by Rusch and Hilsdorf [39]. Hughes

and Chapman [40] confirmed that the concrete did not break at the maximum load, but the softening phenomenon occurred.

From the deformation and failure features, and also the process of microcracks propagation, the stress–strain behavior seems more suitable to be divided into an even damage phase and a local fracture phase bounded by state B, corresponding to the two-stage features in Figure 1a. During the whole process, the nucleation and growth of the random distributed microcracks typically ranges from 0.01 to 1.0 mm in width, leading to a concentration of these microcracks into a narrow zone and producing a visible macroscopic fissure wider than 1.0 mm [41,42]. Before B, the material response could be considered as statistical homogeneous. The density of micro-defects is modest, retaining a dilute degree. After this threshold, the macroscopic response of the specimen strongly depends on the macroscopic main-crack in the FPZ. Local softening and fracture (decrease of the nominal stress with increase of the strain) occurs in the FPZ, while the rest of the region of the specimen enters the unloading process. This portion of the stress–strain curve becomes heavily dependent on the measure gauge length, which could not be treated as a pure material mechanical behavior. Based on the catastrophe theory, the process from damage to fracture for quasi-brittle materials is divided into two phases, globally stable (GS) mode (relevant to the distributed damage accumulation) and evolution induced catastrophic (EIC) mode [43]. The critical state transforming from GS to EIC plays a key role during the whole process, and exhibits the critical sensitivity, with many physics showing abnormal behavior. Experiments on rocks have also shown that there exists a critical value for the fracture of the quasi-brittle material [24,44].

For the location of the critical state B, many experiments [45,46] showed that it lies in the softening region behind the peak state A in a tensile stress–strain curve. However, the accurate location of B is still uncertain, and almost all the theoretical analysis in literatures ignored the identification of the critical state, treating A and B simply as the same one.

## 3. Materials and Methods

### 3.1. Basis of Statistical Damage Theory

It is well known that statistic physics is a ligament that communicates continuum mechanics, damage mechanics, and material mechanics. Statistical damage theory ignores the microscopic details of damage, and the representative volume element (RVE) is abstracted as a complex system composed of $N$ ($N \to \infty$) mesoscopic units (micro-spring, micro-bar, etc., with the same sectional area $dA$ and stiffness $dk$). The heterogeneity of the material is introduced by endowing each unit with different mechanical parameters (strength, characteristic strain, etc.). The macroscopic mechanical properties of concrete depend on the statistical mechanical properties of individual mesoscopic elements. The macroscopic mechanical properties could be described by using a phenomenological model with statistical method.

According to the number of mesoscopic mechanical parameters adopted, statistical damage models could be classified into two types, i.e., the single-parameter model and the double-parameter model, which are represented by the bundle parallel bar system (PBS) [30] and the improved parallel bar system (IPBS) [35,36], respectively.

#### 3.1.1. The Series-Parallel Spring Stochastic Damage Model

Based on the PBS, Li and Zhang [47] presented the series-parallel spring stochastic damage model for concrete under uniaxial tensile test. The core idea is to extend the single-layer parallel element model to multiple layers, which can better simulate the mechanical phenomenon such as localized failure and stress drop. The tensile specimen is abstracted as a complex system composed of n typical units (a unit is composed of numerous micro-springs in parallel with equal spacing) in series. $F$ is the tensile force. $H = nh_0$ is the height of the model, where $h_0$ is the material characteristic height and three times the maximum particle size of aggregate presented by Bažant and Oh [48]. In this model, each micro-spring has the same elastic modulus and cross-sectional area. $\varepsilon_{Ri}$ is the fracture

strain for micro-spring *i*, as a random variable obeying the same probability distribution for each layer unit, where Weibull distribution and Lognormal distribution are often used.

Since the prominent feature of concrete failure is local failure, the failure of the series-parallel spring model is caused by the fracture of a unit body, called the main crack unit body (corresponding to the FPZ), and the others as the non-main crack unit bodies. Define a characteristic strain, namely, the critical strain. Before the critical strain, all unit bodies have the same deformation. After the critical strain, the deformation of the main crack unit body continues to increase, while in the non-main crack unit bodies will occur the unloading behavior with the tensile strain no longer increased. Therefore, the average stress–strain curve after the critical state for the concrete tensile specimen shows the obvious size effect, and stress drop will occur when the specimen size is long enough.

### 3.1.2. The Improved Parallel Bar System (IPBS)

Based on the PBS, the IPBS was proposed by Chen et al. [35] to simulate the damage evolution in the FPZ of quasi-brittle material. The fundamental assumptions for the damage mechanism are as follows: (1) the essence of damage and fracture for a quasi-brittle material is the nucleation, growth, and coalescence of micro-cracks, the stress redistribution and adjustment of the force skeleton in the microstructure. (2) The influence of these irreversible micro-changes on macroscopic mechanical properties of material could be addressed by two dominant aspects: decrease of the effective cross-section area and degradation of the elastic modulus corresponding to the effective bearing position. (3) The two damage modes can be simulated, respectively, by rupture and yield of the micro-bar; and the essence of failure can be interpreted as a continuum accumulation and evolution of the two damage modes.

In this model, each micro-bar is composed of spring, cementation bar, and sliding block. The micro-bar *i* has two feature strains, i.e., the fracture strain $\varepsilon_{Ri}$ and the yield strain $\varepsilon_{yi}$, and it may have two kinds of failure modes (elastic-fracture and yield-fracture) according to the values between $\varepsilon_{Ri}$ and $\varepsilon_{yi}$ (as shown in Figure 2a). By introducing some simplifying assumptions, the two kinds of damage evolution process in the meso-scale are decoupled (further detailed discussion can be referred to from Chen et al. [35]). As shown in Figure 2b, $q(\varepsilon)$ and $p(\varepsilon)$ are the independent probability density functions to $\varepsilon_{Ri}$ and $\varepsilon_{yi}$, respectively; where $\varepsilon_{ymin} = \varepsilon_{Rmin}$ denote the minimums of $\varepsilon_{yi}$ and $\varepsilon_{Ri}$; $\varepsilon_{ymax} < \varepsilon_{Rmax}$ are the corresponding maximums. In the course of damage evolution, the cross-sectional area $A_0$ in the initial state will gradually transform into three parts, i.e., $A_1$, $A_2$, $A_3$, denoting the cross-sectional areas corresponding to the fractured bars, the yielded bars, and the bars in elastic state, respectively. $A_E$ is the effective cross-section area bearing load. It satisfies $A_0 = A_1 + A_2 + A_3$ and $A_E = A_2 + A_3$.

The macroscopic stress–strain curves described by IPBS are drawn in Figure 2c. $\sigma_N$ is the nominal stress corresponding to $A_0$; $\sigma_E$ is the effective stress corresponding to $A_E$; $\overline{\sigma}$ is the elastic stress corresponding to $A_3$. $\varepsilon_{cr} = \varepsilon_{ymax}$; $\varepsilon_u = \varepsilon_{Rmax}$.

By $\varepsilon_{ymax}$, the whole process predicted by the IPBS can be divided into two phases, i.e., partial yield phase ($0 \leq \varepsilon < \varepsilon_{ymax}$) and full yield phase ($\varepsilon_{ymax} \leq \varepsilon \leq \varepsilon_{Rmax}$). When $\varepsilon = \varepsilon_{ymax}$, all the micro-bars in IPBS will yield, and $\sigma_E$ will reach its maximum. After this state, $\sigma_E$ will keep the maximum constant with the further increase of $\varepsilon$. Therefore, we could use the IPBS to simulate the two-stage deformation and failure characteristics of quasi-brittle materials in the FPZ, by assuming that $\varepsilon_{ymax}$ corresponds to the critical state B in Figure 1b.

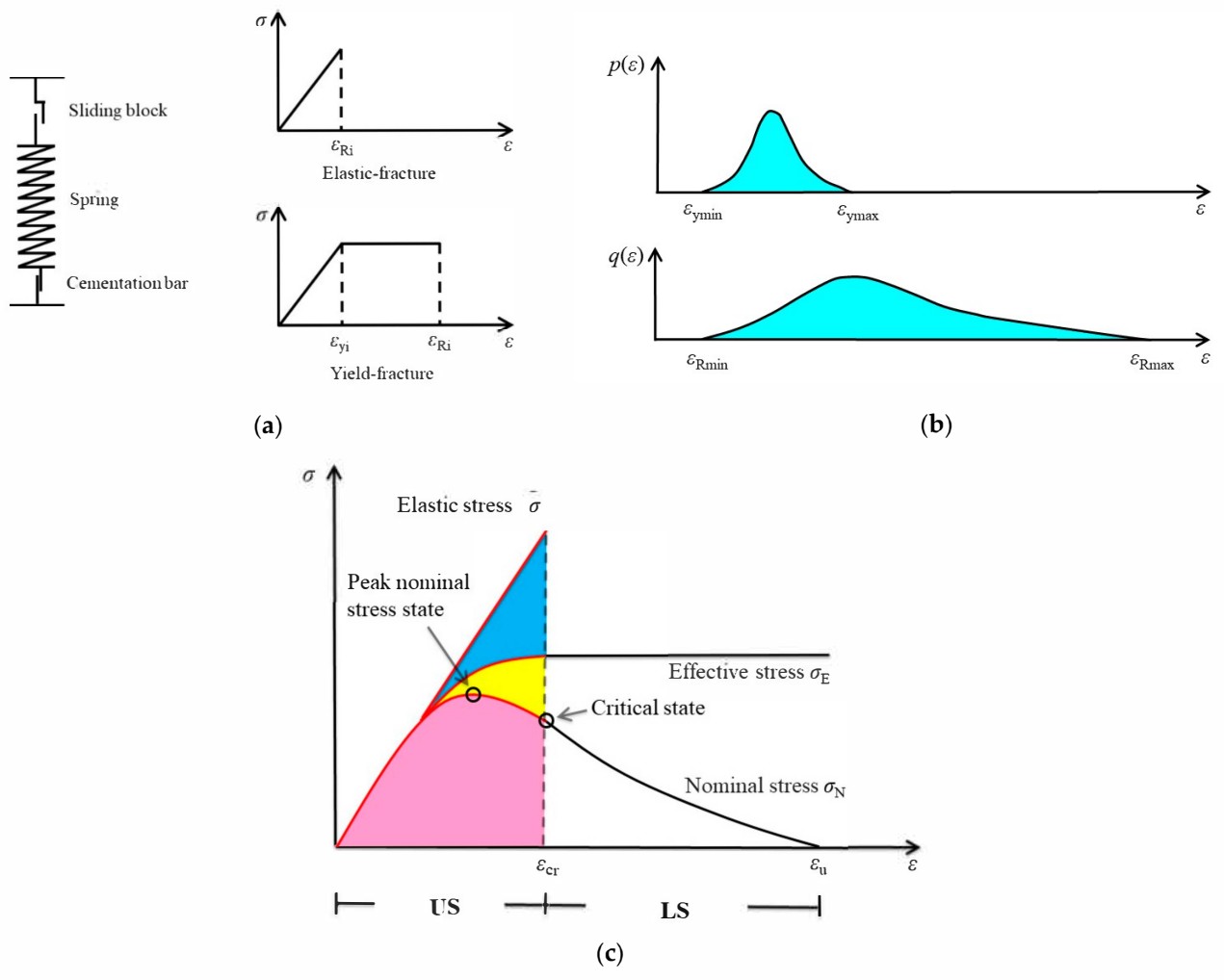

**Figure 2.** Mechanical behavior of concrete under uniaxial tension described by the Improved Parallel Bar System (IPBS):
(**a**) micro bar and failure mode; (**b**) damage evolution process on meso-scale; (**c**) stress–strain curves on macro-scale.

The constitutive relation can be expressed as follows:

(1)  Partial yield phase ($0 < \varepsilon < \varepsilon_{ymax}$)

$$\sigma_N = E_0(1 - D_y)(1 - D_R)\varepsilon \tag{1}$$

$$\sigma_E = E_0(1 - D_y)\varepsilon \tag{2}$$

$$D_y = \int_0^\varepsilon p(\varepsilon)d\varepsilon - \frac{\int_0^\varepsilon p(\varepsilon)\varepsilon d\varepsilon}{\varepsilon} \tag{3}$$

$$D_R = \int_0^\varepsilon q(\varepsilon)d\varepsilon \tag{4}$$

where $D_y$ and $D_R$ denote the accumulated damage variables of elastic modulus of IPBS due to the yield and fracture of the micro-bars; $D_R$ also represents the cumulative distribution function of $q(\varepsilon)$.

(2)  Full yield phase ($\varepsilon_{ymax} \leq \varepsilon < \varepsilon_{Rmax}$)

$$\sigma_N = E_0(1 - D_{ymax})(1 - D_R)\varepsilon_{ymax} \tag{5}$$

$$\sigma_E = E_0(1 - D_{ymax})\varepsilon_{ymax} \tag{6}$$

$$D_{\text{ymax}} = 1 - \frac{\int_0^{\varepsilon_{\text{ymax}}} p(\varepsilon)\varepsilon\mathrm{d}\varepsilon}{\varepsilon_{\text{ymax}}} = \text{constant} \tag{7}$$

$$D_{\text{R}} = \int_0^{\varepsilon_{\text{ymax}}} q(\varepsilon)\mathrm{d}\varepsilon + \int_{\varepsilon_{\text{ymax}}}^{\varepsilon} q(\varepsilon)\mathrm{d}\varepsilon \tag{8}$$

where $D_{\text{ymax}}$ is the value of $D_{\text{y}}$ corresponding to $\varepsilon_{\text{ymax}}$.

IPBS also could simulate the unloading process of the rest of the region of the specimen in the local fracture phase. $\sigma_{\text{N}}$ and $\sigma_{\text{E}}$ can be expressed by Equations (1) and (2). $D_{\text{y}}$ and $D_{\text{R}}$ can be expressed by Equations (9) and (10).

$$D_{\text{y}} = \frac{\int_0^{\frac{\varepsilon_{\text{ymax}}-\varepsilon}{2}} p(\varepsilon)\varepsilon\mathrm{d}\varepsilon}{\varepsilon} + \int_0^{\frac{\varepsilon_{\text{ymax}}-\varepsilon}{2}} p(\varepsilon)\mathrm{d}\varepsilon + \frac{\varepsilon_{\text{ymax}}\int_{\frac{\varepsilon_{\text{ymax}}-\varepsilon}{2}}^{\varepsilon_{\text{ymax}}} p(\varepsilon)\mathrm{d}\varepsilon}{\varepsilon} - \frac{\int_{\frac{\varepsilon_{\text{ymax}}-\varepsilon}{2}}^{\varepsilon_{\text{ymax}}} p(\varepsilon)\varepsilon\mathrm{d}\varepsilon}{\varepsilon} \tag{9}$$

$$D_{\text{R}} = D_{\text{R}}(\varepsilon_{\text{ymax}}) = \int_0^{\varepsilon_{\text{ymax}}} q(\varepsilon)\mathrm{d}\varepsilon = \text{constant} \tag{10}$$

If the length of specimen ($H$) and the length of the FPZ($h_0$) are given, the average stress–strain curve of the specimen during the local fracture phase could be determined by Equations (5) to (10), which would show the obvious size effect [35].

### 3.1.3. Description of the Damage Evolution Process by IPBS

A typical nominal stress–strain curve and a predicted effective stress–strain curve under uniaxial tension are shown in Figure 3a,b. Signs D and E denote the peak nominal stress state and the critical state, respectively. A denotes the limit elastic state, B and C denote two states in strengthen segment, F and F$'$ denote two states in soften segment. The uniaxial tensile damage evolution process of concrete is simulated by using the microscopic spring beam model and the cylinder model, respectively.

In Figure 3c, the micro-bar has two kinds of failure modes, brittle-fracture and yield-fracture. A corresponds to the limit elastic state, where all the micro-bars remain in the elastic state. B, C, D correspond to the two states in strengthen segment and the peak nominal stress state, where some micro-bars are out of work due to fracture, some are yield, and the others still in the elastic state. E corresponds to the critical state, where all the residual bars are yield. F corresponds to the local breach phase in the FPZ, where the yielded bars continue to fracture, exhibiting the softening phenomenon. F$'$ corresponds to the local breach phase in the other region, exhibiting the unloading phenomenon.

In Figure 3d, the tensile specimen is divided into three regions, the elastic region, the yield-damage region, and the rupture-damage region, during the whole damage process. With the growth of damage, the elastic region will gradually transform into the other two kinds of regions. When it reaches the critical state E, the elastic region will disappear, the tensile traction will be fully borne by the yield-damage region, and then the damage of specimen will change into the local breach phase.

In Figure 2c, it shows the stress–strain curves predicted by the IPBS, where $\overline{\sigma}$ is the ideal elastic stress corresponding to the elastic region, $\sigma_{\text{E}}$ is the effective stress corresponding to $A_{\text{E}}$ (the cross-sectional area relevant to the elastic region and the yield-damage region), $\sigma_{\text{N}}$ is the nominal stress corresponding to $A_{\text{N}}$ (the initial total cross-sectional area).

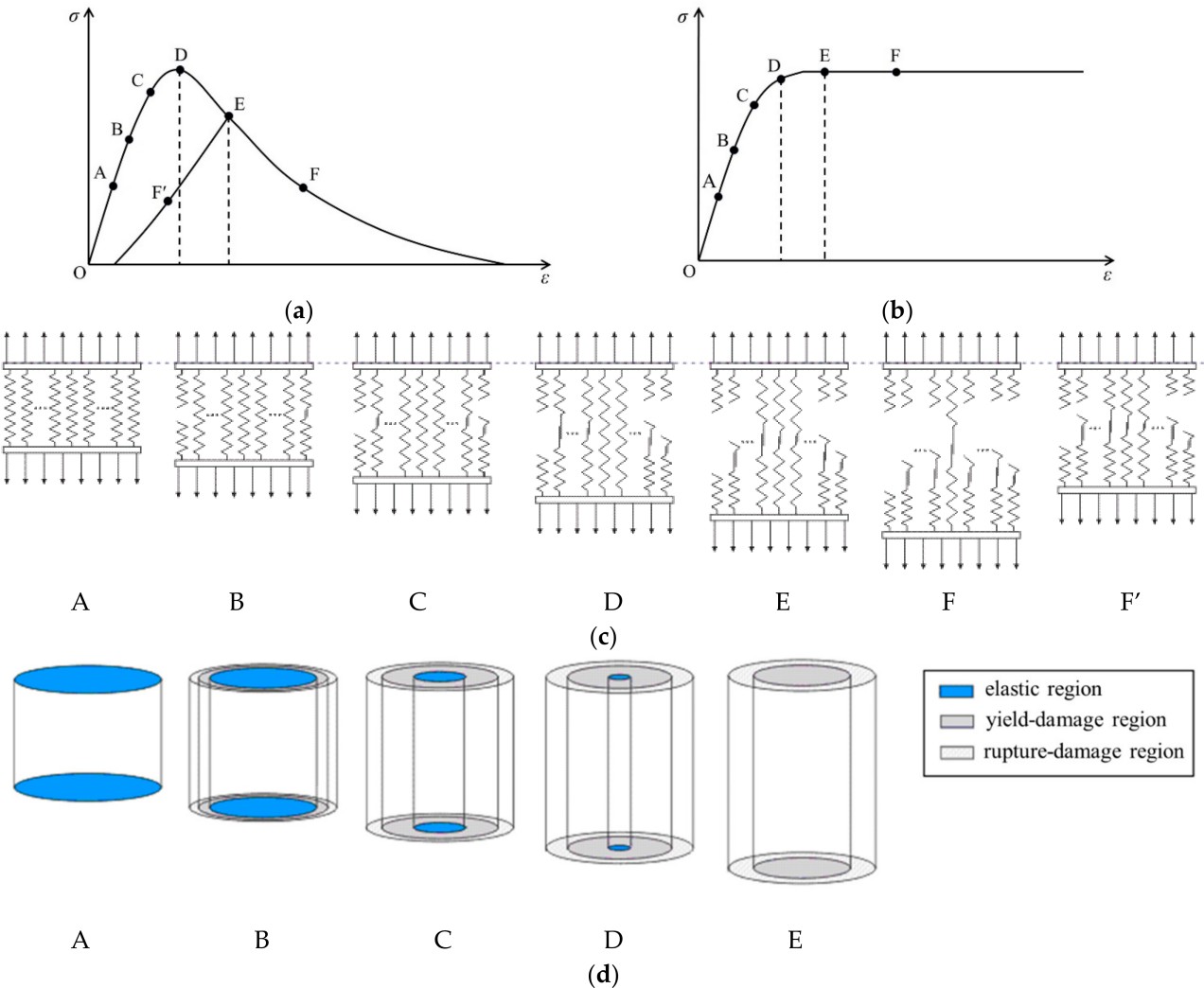

**Figure 3.** Uniaxial tensile damage evolution process: (**a**) nominal stress–strain curve; (**b**) effective stress–strain curve; (**c**) microscopic spring beam model; (**d**) cylinder model.

### 3.1.4. Dialectical Unification between Degeneration and Evolution

Natural dialectics [49] holds that, the process of evolution in nature is the unity of opposites between evolution and degeneration, which exist and occur simultaneously. Evolution-oriented processes often inherently involve degradation, and vice versa. Evolution and degradation are two opposite trends in nature. They are closely combined and inseparable. Each side is the condition for the other side to occur. The combination of the two forms a circular spiral propulsion mode for the evolution of nature, which makes the evolution process of nature appear periodic.

A novel fundamental assumption of mesoscopic damage for concrete material was proposed by Chen et al. [35]. As shown in Figure 4a, it indicated that the damage evolution of concrete materials could be summarized as two kinds of mechanisms on a mesoscopic scale. On the one hand, with the increase of deformation, the initiation, propagation, and coalescence of micro-cracks and micro-defects, as well as the acoustic emission phenomenon, will occur randomly in the microstructure, which is the so-called degradation effect. On the other hand, due to the initiation and propagation of microcracks, stress redistribution and adjustment of force skeleton will take place in the microstructure. We may be able to understand the second mechanism as the active adjustment of the material system itself. In this way, the force skeleton of the microstructure is further adjusted and optimized, and the potential mechanical ability of the material is further developed to

bear more external loads (effective stress). Therefore, the second type of mechanism is called the strengthening effect here. Berthier et al. [24] indicated the quasi-brittle failure emerges from the interaction between the elements constituting the material. They also highlighted the central role played by the mechanism of load redistribution to control the failure behavior of quasi-brittle solids.

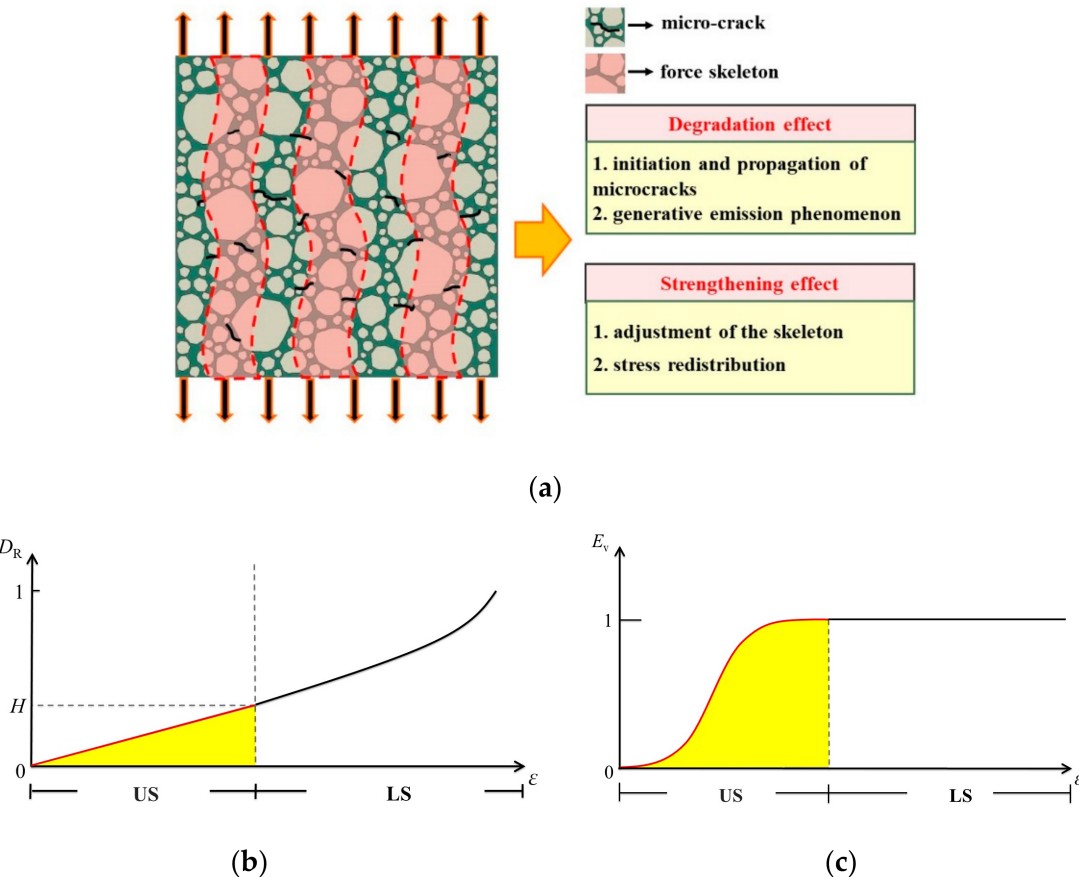

**Figure 4.** Two types of mesoscopic mechanism of quasi-brittle material: (**a**) damage evolution mechanism on a mesoscopic scale; (**b**) degradation factor; (**c**) evolutionary factor.

According to the assumption, the deformation and failure of concrete material is not only the process of "deterioration" of macroscopic and microscopic mechanical properties, but also the process of "evolution" in which the material goes through their own active-adjustment to adapt to the external load environment. When the inner adjustment capabilities of the material exert to its limit, the damage evolution of the material will switch from the statistical uniform damage to the local breach. This local failure process generally takes place in the weakest region (such as the fracture process zone in concrete).

The above viewpoints are consistent with the catastrophe theory [43], which holds that the failure process of solid materials could be divided into two phases, GS and EIC. The critical state transforming from GS to EIC plays a key role during the whole process, and exhibits the critical sensitivity. The two-stages embody the process from quantitative change to qualitative change.

Figure 4b,c shows the two mechanisms of the deformation and failure of concrete on a mesoscopic scale, i.e., the "degradation" and "strengthening", respectively.

As shown in Figure 4c, $E_v$ is introduced as the evolutionary factor to describe the "strengthening" process, corresponding to the yield damage mode. The expression is as follows:

$$E_v = \int_0^\varepsilon p(\varepsilon)d\varepsilon \quad (0 \le \varepsilon \le \varepsilon_{ymax}) \tag{11}$$

$E_v$ could be used to assess the extent to which the potential mechanical capabilities (adjustment capabilities of force skeleton in microstructure) of materials are developed, ranging from 0 to 1. When $E_v = 0$, it corresponds to the initial undamaged state. When $E_v = 1$, it corresponds to the critical state, at which the potential adjustment capabilities of materials reach their limits, $\sigma_E$ reaches its maximum value, and then the materials enter into the local catastrophic stage. The whole process embodies the characteristics of "quantum" to "qualitative", in which the yield damage mode plays a key role.

As shown in Figure 4b, $D_R$ is as the fracture damage variable, and could be used to describe the "degradation" processes. It has the similar physical meanings with $D$ defined by Dougill [50], which characterizes the initiation and propagation of microcracks and leads to the reduction of the effective stress area. At the uniform damage stage, it ranges from 0 to H (a certain smaller value), and it seems not to be playing a key role in the whole process.

### 3.2. Statistical Damage Model for FRC in Uniaxial Tension

3.2.1. Influence Mechanism of Fiber

A large number of experimental studies [1–9,12] have shown that, the influencing factors mainly include: the strength of the matrix concrete, the type of fiber, the length diameter ratio and the volume ratio of fiber, the bonding strength between fiber and matrix, and the distribution and orientation of the fiber in the matrix. The existing theories mainly focus on fiber and matrix, and study the interaction between single fiber and matrix.

(1)   Fiber spacing theory

Romualdi and Batson [13] analyzed the limitation and restraint mechanism of steel fiber to concrete crack, and put forward the theory of fiber spacing to explain the mechanism of fracture enhancement of the SFRC. This theory holds that the existence of fiber could significantly reduce the size and quantity of micro-cracks, reduce the stress concentration degree of the crack tip, and restrain the occurrence and expansion of micro-cracks. The key to fiber reinforcement is the average spacing of the fibers. With the increase of average spacing, the fiber's ability to restrain the crack initiation and expansion will be greater, and the strength of the SFRC will be higher. The size of the average spacing of the fibers depends on the number of active fibers in the volume of the unit matrix.

(2)   Reinforcement rules for composite materials

Swamy [14] proposed the reinforcement rules for composite materials based on the mechanics principle of composite materials. The SFRC is simplified as fiber and concrete two-phase composite materials, and the properties of the composite material are cumulative for each phase. Due to the non-homogeneity of ordinary concrete structure, irregular stress concentration would occur within the matrix when the structure is pulled. When the ultimate tensile strength of ordinary concrete is less than the tensile stress of the stress concentration point, the stress concentration point will create cracks. Since the tensile strength of the steel fibers is much higher than the tensile strength of the concrete matrix, the incorporation of steel fiber can effectively inhibit and delay the initiation and expansion of micro-cracks in ordinary concrete.

In other words, these randomly distributed fibers can effectively prevent the expansion of micro-cracks in concrete and delay the formation of macro-cracks. Hence, it can significantly improve the macroscopic mechanical properties of concrete, such as tensile, bending, impact, fatigue resistance, and so on. The main factors that influence fiber behavior include: the species, geometric features, and content of the fibers, the bonding properties of the fibers to the concrete matrix, distribution and orientation of the fibers in matrix. Most of the existing constitutive models for FRC are adopted from the macroscopic phenomenological mathematical expressions (as shown in Table 1), of which parameters lack a clear physical meaning. Therefore, it is difficult for them to reflect the influence of fiber content on the mesoscopic damage evolution of the concrete.

**Table 1.** Theoretical stress–strain models of SFRC under tension in literature.

| Origin of Data | Tensile Strength/MPa | Fiber Types | Main Formula |
|---|---|---|---|
| Gao [20] | 2~3 | Melt-extracted | $\begin{cases} y = A + (3 - 2A)x + (A - 2)x^2 & (x \leq 1) \\ y = \frac{x}{a(x-1)^{1.7}+x} & (x > 1) \end{cases}$ |
| Han et al. [22] | 3~5 | Large steel fiber | $\begin{cases} y = 1.2x - 0.2x^6 & (x \leq 1) \\ y = \frac{x}{a_{ft}(x-1)^{1.7}+x} & (x > 1) \end{cases}$ |

### 3.2.2. Practical Expressions of the IPBS

According to the statistical damage theory mentioned above, the macroscopic mechanical behavior (nominal/effective stress–strain curve) of concrete under uniaxial tension, is determined by the cumulative evolution process of the fracture and yield damage modes in a meso-scale, as shown in Figure 5. The whole process includes the homogeneous damage accumulation stage and local failure stage. Two characteristic states are distinguished, namely peak nominal stress state and critical state. $\varepsilon_{cr}$ and $\varepsilon_{u}$ denote the strains of critical state and ultimate state in the nominal stress–strain curve.

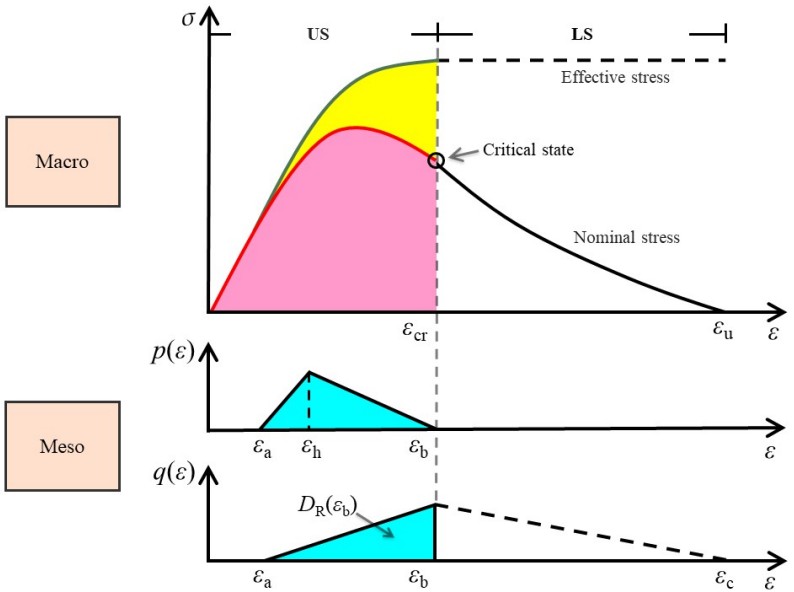

**Figure 5.** Relationship between constitutive behavior on macro-scale and damage evolution process on meso-scale.

As the probability density functions corresponding to fracture and yield damage in the meso-scale respectively, $q(\varepsilon)$ and $p(\varepsilon)$, may be subject to complex statistical distribution laws in the true case, such as Weibull, Normal, and other distributions. Considering the complexity of the problem, $q(\varepsilon)$ and $p(\varepsilon)$ are both assumed to obey the independent triangle distribution form in a specific calculation, as shown in Figure 5. Analyses [35–38] show that the true stress–strain test curves could be well fitted and the evolution mechanism of non-homogeneous damage on the meso-scale could be well revealed, when $q(\varepsilon)$ and $p(\varepsilon)$ are adopted by the simplified triangular distributions. $\varepsilon_{a}$ is the initial damage strain. $\varepsilon_{h}$ is the strain corresponding to the peak value of $p(\varepsilon)$. $\varepsilon_{b}$ is the strain corresponding to the maximum yield damage state, and also to the peak value of $q(\varepsilon)$. $\varepsilon_{c}$ is the strain corresponding to the maximum fracture damage state. It satisfies $\varepsilon_{b} = \varepsilon_{cr}$ and $\varepsilon_{c} = \varepsilon_{u}$ in uniaxial tension.

Due to the softening segment of the nominal stress–strain curve corresponding to the local failure stage having obvious size effect, the critical state is suggested as the final failure point of the constitutive model in this paper. Hence, the following content in the paper only discusses the constitutive behavior of concrete in the homogeneous damage stage.

At the homogeneous damage stage, $\sigma_N$ and $\sigma_E$ can be obtained by Equations (1)–(4), where $q(\varepsilon)$ and $p(\varepsilon)$ can be expressed as the following:

$$q(\varepsilon) = \begin{cases} 0 & (\varepsilon \leq \varepsilon_a) \\ \dfrac{2H(\varepsilon - \varepsilon_a)}{(\varepsilon_b - \varepsilon_a)^2} & (\varepsilon_a < \varepsilon \leq \varepsilon_b) \end{cases} \tag{12}$$

$$p(\varepsilon) = \begin{cases} 0 & (\varepsilon \leq \varepsilon_a) \\ \dfrac{2(\varepsilon - \varepsilon_a)}{(\varepsilon_h - \varepsilon_a)(\varepsilon_b - \varepsilon_a)} & (\varepsilon_a < \varepsilon \leq \varepsilon_h) \\ \dfrac{2(\varepsilon_b - \varepsilon)}{(\varepsilon_b - \varepsilon_h)(\varepsilon_b - \varepsilon_a)} & (\varepsilon_h < \varepsilon \leq \varepsilon_b) \end{cases} \tag{13}$$

where $H = D_R(\varepsilon_b)$ is the fracture damage value relevant to $\varepsilon_b$.

Define $S$ as the energy absorbing capability, which represents the energy absorbed by concrete in the process of stress and deformation. The expression is as follows:

$$S = \int_0^\varepsilon \sigma_N d\varepsilon \tag{14}$$

$$S_p = \int_0^{\varepsilon_p} \sigma_N d\varepsilon \tag{15}$$

$$S_{cr} = \int_0^{\varepsilon_{cr}} \sigma_N d\varepsilon \tag{16}$$

where $S_p$ and $S_{cr}$ are the energy absorption capacity corresponding to peak nominal stress state and critical state, respectively.

### 3.2.3. Influence of the Fibers on Mesoscopic Damage Mechanism

Statistical damage theory suggests that the change on the macroscopic nonlinear stress–strain behavior of concrete under a complex environment is essentially caused by the microscopic damage mechanism. The fiber reinforcement effect changes the composition and characteristics of the concrete microstructures, the nucleation and growth of microcracks, and the cumulative evolution process of damage, which finally lead to the change on the macro-nonlinear stress–strain behavior of concrete. The above effects can be summarized into two aspects: (1) changes on the composition and mechanical properties of microstructure, measured by $E_0$; (2) changes on the pattern and law of the initiation and propagation of microcracks; in other words, changes on the cumulative evolution process of the two meso-damage modes (yield and fracture), measured by $\varepsilon_a$, $\varepsilon_h$, $\varepsilon_b$, and $H$, which determine the shape of the triangle probability distribution of $q(\varepsilon)$ and $p(\varepsilon)$.

As shown in Figure 6, it is assumed that the microstructure characteristics and meso-damage evolution process of concrete under different fiber contents obey a certain regularity, and the above changes at the meso-level lead to the strengthening of the macroscopic constitutive behavior of concrete. Make $E_0$, $\varepsilon_a$, $\varepsilon_h$, $\varepsilon_b$, and $H$ as a function of fiber volume fraction $\rho$ (%) (here, does not consider the influence of other factors, such as fiber type and geometric features), the expression is as follows:

$$\begin{cases} E_0 = f_1(\rho) \\ \varepsilon_a = f_2(\rho) \\ \varepsilon_h = f_3(\rho) \\ \varepsilon_b = f_4(\rho) \\ H = f_5(\rho) \end{cases} \tag{17}$$

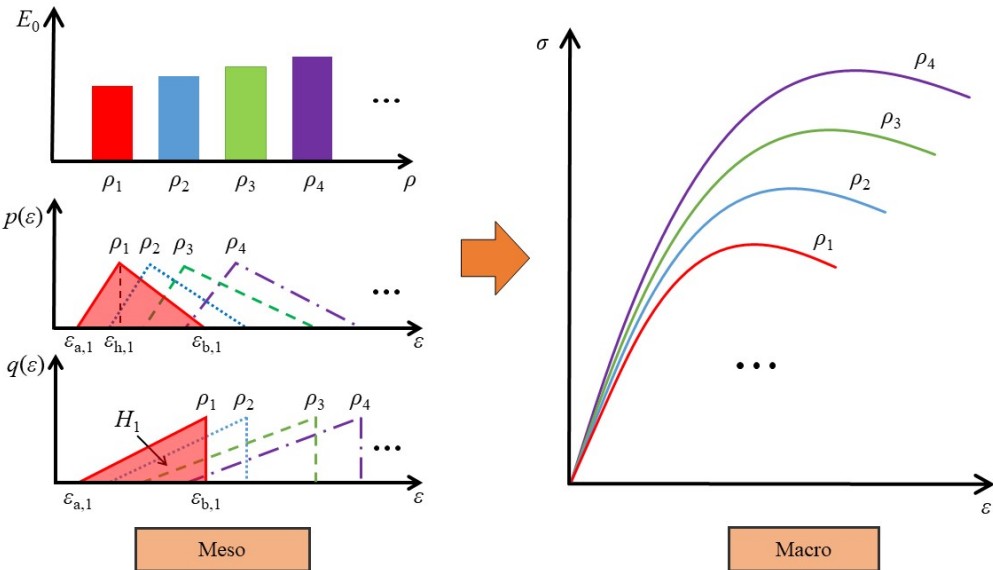

**Figure 6.** The influence of the fiber volume fraction on the damage mechanism of concrete.

As mentioned above, according to these five parameters, we can determine the uniaxial tensile stress–strain behaviors of concrete with different fiber content, and also the laws of the evolution of the mesoscopic damage mechanism.

### 3.2.4. Determination of Model Parameters

Five parameters need to be determined for each nominal stress–strain curve under uniaxial tension, they are $E_0$, $\varepsilon_a$, $\varepsilon_h$, $\varepsilon_b$, and $H$. $E_0$ can be obtained directly from the test curve, taken as the secant modulus from 0.2–0.4 times of peak stress point in the ascending part of the curve to the original point. $\varepsilon_a$, $\varepsilon_h$, $\varepsilon_b$, and $H$ can be obtained by the multivariate regression analysis of the genetic algorithm module in the Matlab toolbox. The solution procedure is summarized as follows:

(1) Create a fitness function including $\varepsilon_a$, $\varepsilon_h$, $\varepsilon_b$, and $H$, and take the sum of the squares of the difference between predicted value and measured value of nominal stress as the optimization criterion.
(2) Initially, set the search interval for the values of the four parameters.
(3) To perform the genetic algorithm, and obtain the optimal solution of the 4 parameters calculated by this iteration. Adjust or narrow the search interval of parameters according to the results.
(4) Repeat step (3), until the optimal solution is obtained.

## 4. Results and Discussion

The presented model is validated by using two sets of uniaxial tensile tests for steel fiber-reinforced concrete with different fiber contents reported by Han et al. [22] and Gao [20]. The relevant experimental information are summarized in Table 2. The rationality and applicability of the presented model are verified, and the damage evolution mechanism of steel fiber concrete in uniaxial tension is discussed.

**Table 2.** Summary of uniaxial tensile test information.

| | | Types | Length to diameter ratio | Volume fraction /% | Equivalent diameter /mm | Average length /mm |
|---|---|---|---|---|---|---|
| Gao [20] | Fiber | Melt-drawn | 50 | 0.5, 1.0, 1.5, 2.0 | 0.5 | 25 |
| | Mixture/kg/m³ | Cement | Water | Sand | Stone | Water reducer |
| | | 450 | 225 | 887.5 | 887.5 | 0 |
| Han et al. [22] | Fiber | Types | Length to diameter ratio | Volume fraction /% | Equivalent diameter /mm | Average length /mm |
| | | Large-end | 44.34 | 0, 0.5, 1.0, 1.5, 2.0, 2.5 | 0.698 | 30.96 |
| | Mixture/kg/m³ | Cement | Water | Sand | Stone | Water reducer |
| | | 450 | 158 | 737 | 1105 | 4.5 |

### 4.1. Comparison with the Test by Han et al., 2006

Han et al. [22] conducted a uniaxial tensile test for a steel fiber-reinforced concrete specimen. The specimen is of dumbbell shape with a total length of 450 mm. The length of the middle tensile region is 170 mm, with a cross section of 100 mm × 100 mm. A large-end steel fiber with a length-to-diameter ratio of 44.34 is adopted. The volume fractions of the steel fiber $\rho$ are 0%, 0.5%, 1.0%, 1.5%, 2.0%, and 2.5%. The theoretical nominal stress–strain curves of steel fiber concrete with different fiber contents calculated by the presented model, corresponding to the homogeneous damage phase, are shown in Figure 7a (the curve with a fiber volume fraction of 1.5% is removed due to the dispersion of the experimental data.). The curves include the ascending and partial descending segments of the stress–strain behavior of the fiber-reinforced concrete. They show a good fitting effect compared to the test data. The predicted effective stress–strain curves are shown in Figure 7b. The relevant calculation parameters are listed in Table 3, where $R^2$ is the correlation coefficient. For this proposed model, the entire process of deformation and failure of concrete under uniaxial tension is understood from the viewpoint of effective stress. In the uniform damage stage, the nominal stress $\sigma_N$ first increases and then decreases, involving with a peak nominal stress state (so-called strength state). The effective stress $\sigma_E$ increases monotonously and reaches its maximum at the critical state. After the critical state, the specimen enters the local failure stage characterized by macroscopic crack propagation. The 3D envelopes of $\sigma_N$-$\varepsilon$ and $\sigma_E$-$\varepsilon$ curves predicted by the presented model are shown in Figure 7c,d. They clearly show the variation trend of the curves of concrete with different fiber contents, and the shape of the curves show a good similarity rule, especially the connection part between the ascending and descending segments. With the increase of steel fiber volume ratio $\rho$ from 0% to 2.5%, the values of $\sigma_N$ and $\varepsilon$ corresponding to the peak nominal stress state and the values of $\sigma_E$ and $\varepsilon$ corresponding to the critical state increase significantly.

The relationship curves of $\rho$-$\varepsilon_a$, $\varepsilon_h$, $\varepsilon_b$ are shown in Figure 8a, which represent the evolution law of the yield damage mode on a mesoscale. The curve of $H$-$\rho$ is shown in Figure 8b, which depicts the evolution law of the fracture damage mode on a mesoscale. The relationship curve of $E_0$-$\rho$ is shown in Figure 8c. The aforementioned five parameters show an obvious regularity with the increase in steel fiber content, in which $\varepsilon_h$, $\varepsilon_b$, and $H$ increase linearly, whereas $E_0$ decreases linearly. The fitting curves and fitting expressions are also shown in the figures.

Figure 9a shows the curves of the evolution process of fracture damage variable $D_R$ with different steel fiber contents, respectively. $D_R$ is closely related to microcrack density, and its evolution process under uniaxial tension is significantly delayed with the increase in fiber content from 0% to 2.5%, which is consistent with the physical background in the microstructure.

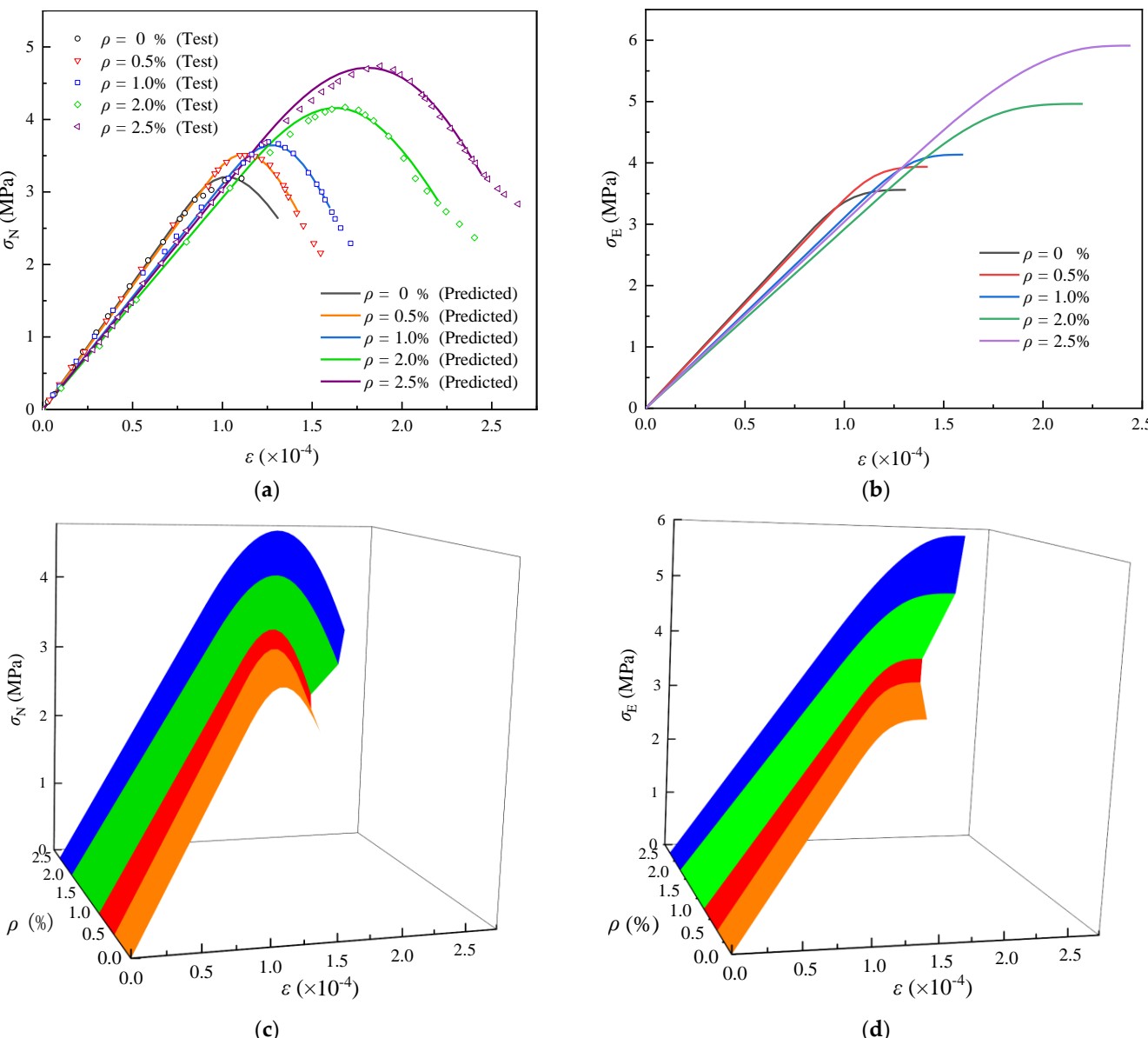

**Figure 7.** Stress–strain curves under uniaxial tension: (**a**) nominal stress–strain curve (Plan view); (**b**) effective stress–strain curve (Plan view); (**c**) envelope of the nominal stress–strain curve (3D view); (**d**) envelope of the effective stress–strain curve (3D view).

**Table 3.** Results for calculation parameter.

| $\rho$ (%) | $E_0/\times 10$ GPa | $\varepsilon_a/\times 10^{-4}$ | $\varepsilon_h/\times 10^{-4}$ | $\varepsilon_b/\times 10^{-4}$ | $H$ | $R^2$ |
|---|---|---|---|---|---|---|
| 0 | 3.481 | 0.771 | 0.993 | 1.311 | 0.260 | 0.9995 |
| 0.5 | 3.413 | 0.909 | 1.134 | 1.427 | 0.307 | 0.9997 |
| 1.0 | 3.107 | 1.032 | 1.362 | 1.598 | 0.327 | 0.9993 |
| 2.0 | 2.921 | 1.127 | 1.771 | 2.210 | 0.419 | 0.9991 |
| 2.5 | 3.046 | 1.181 | 2.211 | 2.441 | 0.450 | 0.9994 |

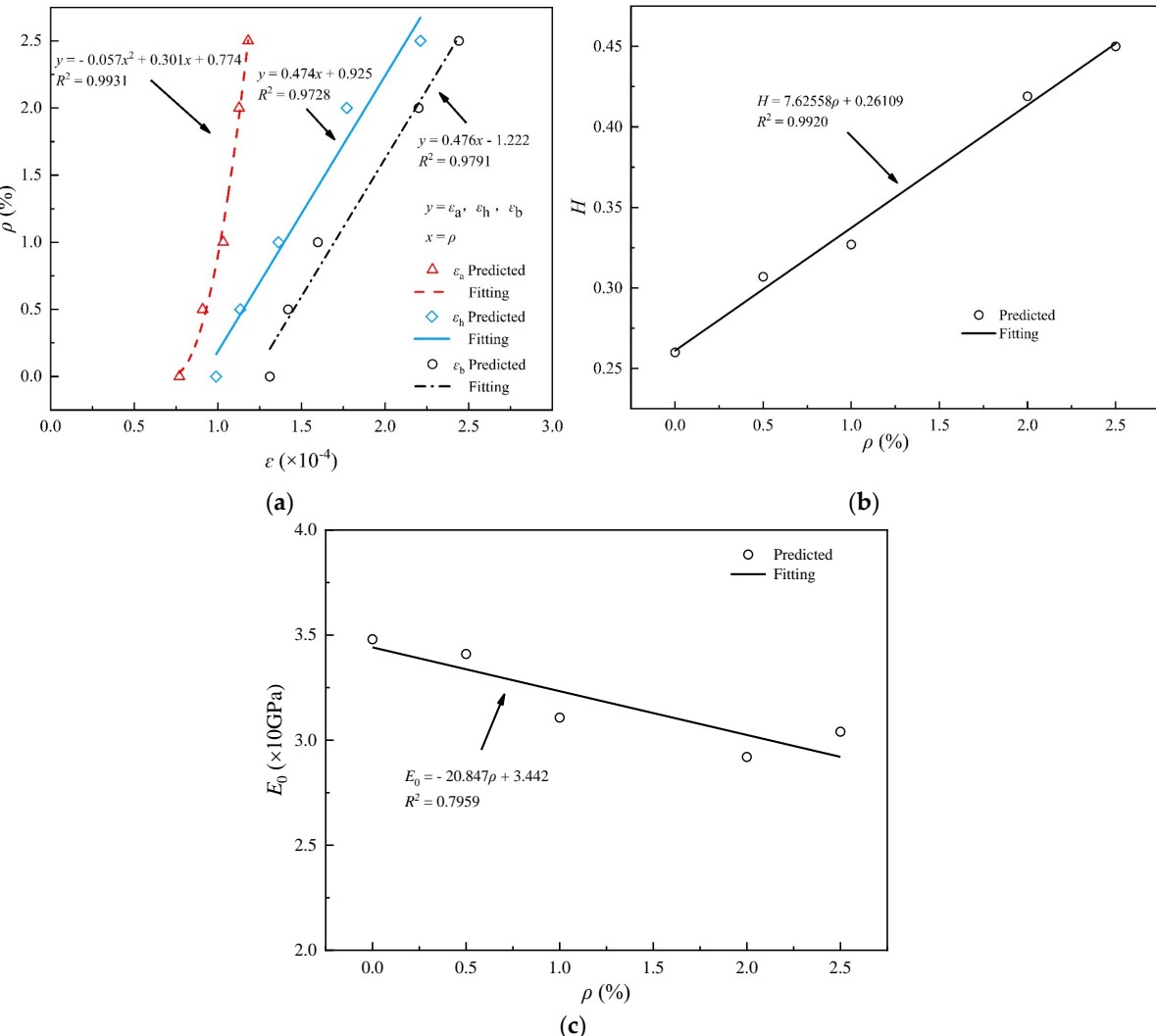

**Figure 8.** Influence of fiber volume fraction on the characteristic parameters: (**a**) $\rho$-$\varepsilon_a$, $\varepsilon_h$, $\varepsilon_b$ curves; (**b**) $H$-$\rho$ curve; (**c**) $E_0$-$\rho$ curve.

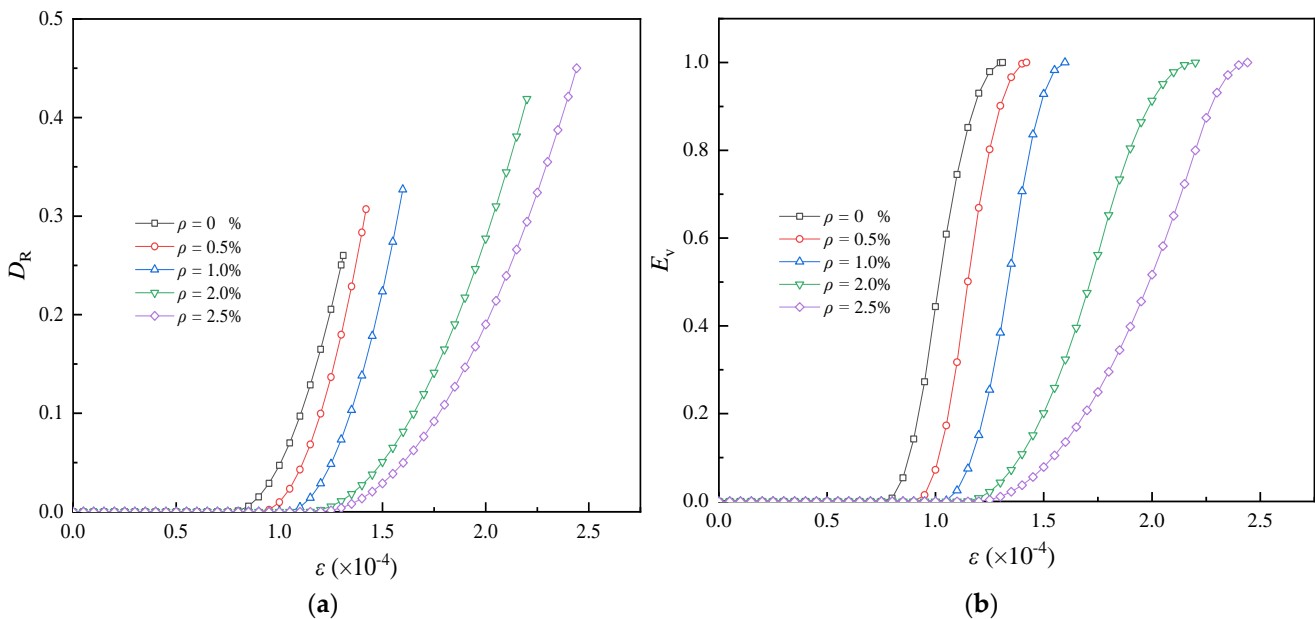

**Figure 9.** Evolution curves of damage variable and evolution factor: (**a**) $D_R$-$\varepsilon$ curves; (**b**) $E_v$-$\varepsilon$ curves.

In Figure 9b, evolutionary factor $E_v$ represents the exerting degree of the underlying mechanical capacity of materials, and its evolution process is delayed with the increase in fiber content. When the critical state, $E_v = 1$, is reached, all the microbars in IPBS will yield and cannot endure much effective stress, then the concrete specimen enters the local failure stage characterized by macroscopic crack growth.

The physical meanings of the above-mentioned characteristic parameters are clear. The prediction results can be used to explore the internal relations among the physical background, the mesodamage evolution mechanism, and the macro-nonlinear mechanical behavior of concrete for adding steel fibers with different contents.

(1)　Damage mechanism on a mesoscale

Local stress concentration will occur in the concrete matrix under tensile load due to the heterogeneity in the microstructure, which will lead to the initiation and expansion of microcracks. After adding steel fiber to the concrete matrix, the composition and mechanical properties of the microstructure will change, and the initiation and penetration of microcracks in the concrete matrix will be significantly inhibited and delayed. The evolution and accumulation process of the mesodamage modes will consequently change. For the yield damage mode, with the increase in $\rho$ from 0% to 2.5%, $\varepsilon_a$ increases from $0.771 \times 10^{-4}$ to $1.181 \times 10^{-4}$, $\varepsilon_h$ increases linearly from $0.993 \times 10^{-4}$ to $2.211 \times 10^{-4}$, and $\varepsilon_b$ increases linearly from $1.311 \times 10^{-4}$ to $2.441 \times 10^{-4}$. For the fracture damage mode, $H$ increases linearly from $0.260 \times 10^{-4}$ at 0% to $0.450 \times 10^{-4}$ at 2.5%. The above-mentioned four parameters can be used to determine the shape of triangular probability distribution corresponding to the mesoscopic yield and fracture damage evolution process, which can demonstrate intuitionistic physical pictures to people to understand the entire process on a mesoscale.

(2)　Mechanical behavior in macroscale

The macroscopic nonlinear stress–strain behavior of concrete is determined by the mechanical properties of the microstructure and the evolution of mesodamage. After adding steel fiber to the concrete matrix, with $\rho$ varying from 0% to 2.5%, the composition and mechanical properties of the microstructure will change, which will lead the initial elastic modulus $E_0$ to decrease from $3.481 \times 10$ GPa to $3.046 \times 10$ GPa. The nominal stress–strain curves generally show "strengthened" features due to the change in the mesodamage evolution process characterized by the four parameters ($\varepsilon_a$, $\varepsilon_h$, $\varepsilon_b$, and $H$) with the increase in fiber content. The nominal stresses corresponding to the peak nominal stress state and the critical state, $\sigma_{N,p}$ and $\sigma_{N,cr}$, increase from 3.134 and 2.612 MPa to 4.713 and 3.251 MPa, respectively. The tensile strains corresponding to the peak nominal stress state and the critical state, $\varepsilon_{N,p}$ and $\varepsilon_{N,cr}$, increase from $1.051 \times 10^{-4}$ and $1.303 \times 10^{-4}$ to $1.801 \times 10^{-4}$ and $2.442 \times 10^{-4}$, respectively. No significant change is observed in the slope of the descending branch of the nominal stress–strain curve, and the ductility performance is significantly improved.

Figure 10 shows the change curves of energy absorption capacity S with fiber volume fraction, where $S_p$ and $S_{cr}$ represent the energy absorption capacities corresponding to the peak nominal stress state and the critical state, respectively. With the increase in fiber content $\rho$ from 0% to 2.5%, $S_p$ increases from 0.172 to 0.479 kPa, $S_{cr}$ increases from 0.266 to 0.749 kPa, and the difference between $S_{cr}$ and $S_p$ increases from 0.094 to 0.270 kPa. In previous studies, $S_p$ was commonly used to characterize the energy absorption capacity of concrete before failure. In the present study, $S_{cr}$ is suggested to characterize the energy absorption capacity of concrete before failure. This approach allows full consideration of the potential mechanical properties of the material.

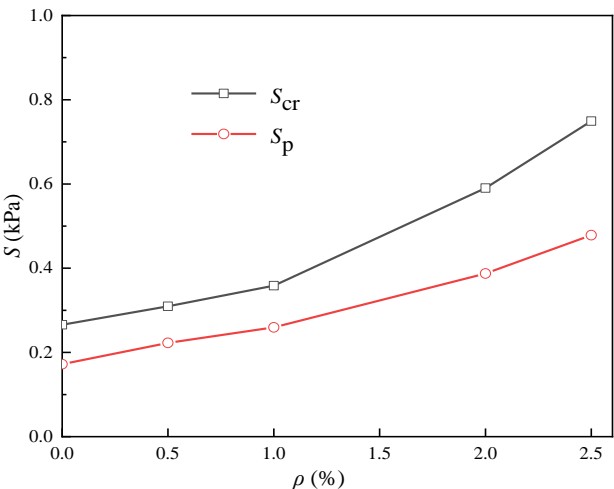

**Figure 10.** Influence of fiber volume fraction on the energy absorption capacity.

*4.2. Comparison with the Test by Gao, 1991*

Gao [20] also conducted a uniaxial tensile test for steel fiber-reinforced concrete specimen. The rectangular block with size of 100 mm × 100 mm × 500 mm is used in the experiment. Deformed steel bars with a diameter of 20 mm are inserted into both ends of the specimen. Fiber type is the melt-drawn steel fiber with length-to-diameter ratio of 50. The volume fractions of the steel fiber $\rho$ are 0.5%, 1.0%, 1.5%, and 2.0%.

As shown in Figure 11a, the predicted nominal stress–strain curves relevant to the homogeneous damage phase agree well with the test curves. The predicted effective stress–strain curves are shown in Figure 11b. The calculation parameters are listed in Table 4. The 3-D envelopes of $\sigma_N$-$\varepsilon$ and $\sigma_E$-$\varepsilon$ curves predicted are shown in Figure 11c,d, in order to better show the variation trend of the curves. It clearly shows that the shape of the curves has obvious similarity law with the increase in the volume ratio of steel fiber. With the increase of $\rho$ from 0.5% to 2.0%, the stresses and strains relevant to the peak nominal stress state and the critical state increase significantly. The change of the slope of the descending section of the nominal stress–strain curves is not obvious, meanwhile the ductility is further improved.

Figure 12a shows the relationship curves of $\rho$-$\varepsilon_a$, $\varepsilon_h$, $\varepsilon_b$. With $\rho$ ranging from 0.5% to 2.0%, $\varepsilon_a$ decreases linearly from $0.394 \times 10^{-4}$ to $0.194 \times 10^{-4}$, $\varepsilon_h$ decreases linearly from $0.164 \times 10^{-4}$ to $0.030 \times 10^{-4}$, meanwhile $\varepsilon_b$ increases linearly from $1.601 \times 10^{-4}$ to $2.411 \times 10^{-4}$. Figure 12b shows the relationship curves of $H$-$\rho$, $H$ decreases linearly from 0.202 at 0.5% to 0.162 at 2.0%. Figure 12c shows the change curve of $E_0$ with $\rho$, which has no obvious changes. The fitting curves and fitting functions of the above five parameters are also shown in the figures.

Figure 13a shows the evolution curves of $D_R$, which characterize the cumulative evolution process of the fracture damage mode of concrete on a mesoscale, respectively. With the increase of $\rho$ from 0.5% to 2.0%, the evolution process of $D_R$ has been significantly delayed.

Figure 13b shows the evolution curves of $E_v$, which characterizes the release process of potential mechanical capacity of concrete. Its evolution process has accelerated in early and then obviously delayed in later, with the increase of steel fiber content. When it reaches the critical state, $E_v = 1$. It indicates that the potential mechanical capacity of the material to be fully released, and then the concrete specimen enters into the local failure stage. The whole process embodies the conversion from quantitative change to qualitative change.

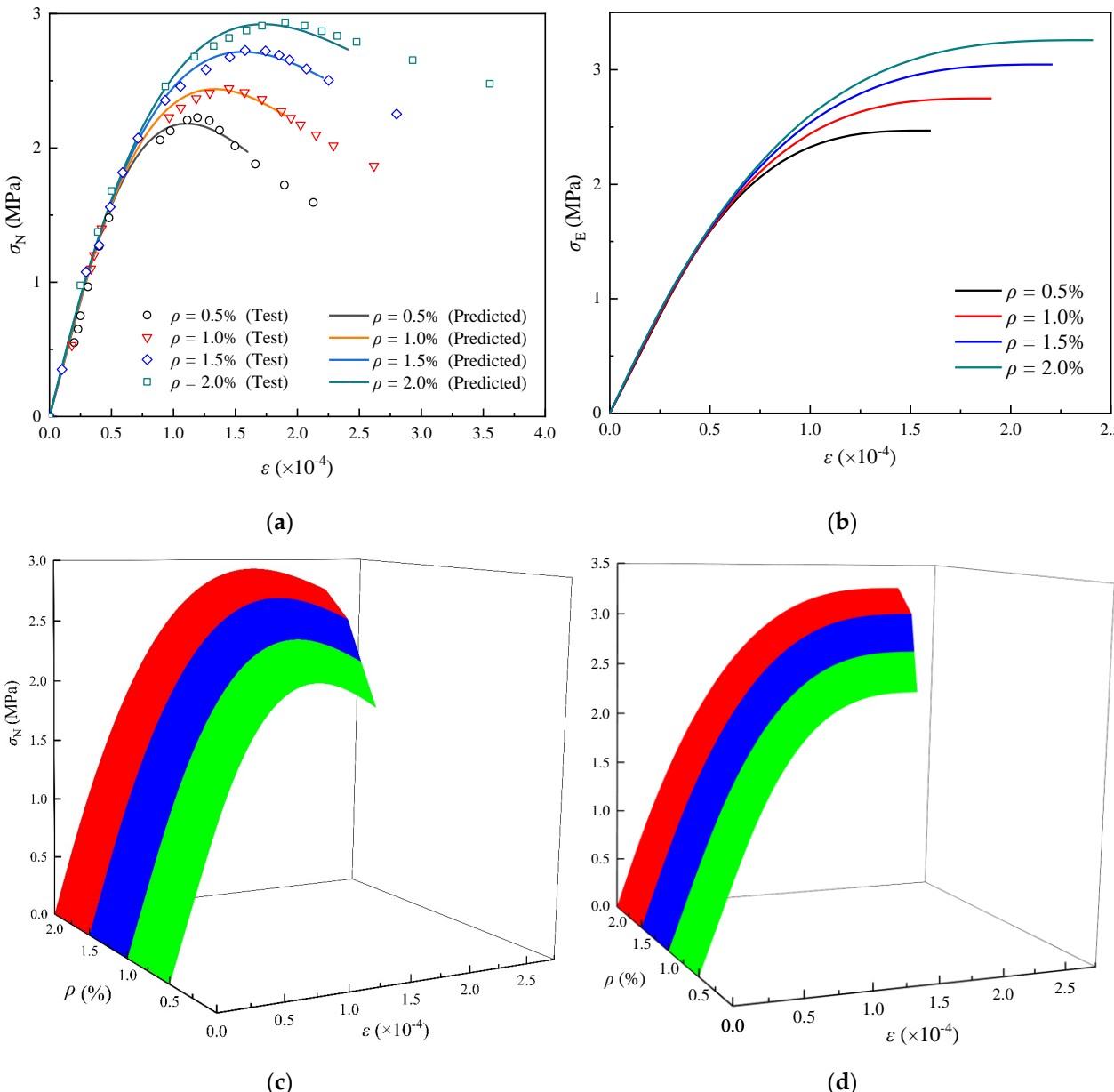

**Figure 11.** Stress–strain curves under uniaxial tension: (**a**) nominal stress–strain curve (Plan view); (**b**) effective stress–strain curve (Plan view); (**c**) envelope of the nominal stress–strain curve (3D view); (**d**) envelope of the effective stress–strain curve (3D view).

**Table 4.** Results for calculation parameter.

| $\rho$ (%) | $E_0/\times 10$ GPa | $\varepsilon_a/\times 10^{-4}$ | $\varepsilon_h/\times 10^{-4}$ | $\varepsilon_b/\times 10^{-4}$ | $H$ | $R^2$ |
|---|---|---|---|---|---|---|
| 0.5 | 3.427 | 0.164 | 0.394 | 1.601 | 0.202 | 0.9773 |
| 1.0 | 3.512 | 0.088 | 0.357 | 1.905 | 0.186 | 0.9692 |
| 1.5 | 3.649 | 0.033 | 0.261 | 2.209 | 0.172 | 0.9846 |
| 2.0 | 3.712 | 0.030 | 0.194 | 2.411 | 0.162 | 0.9721 |

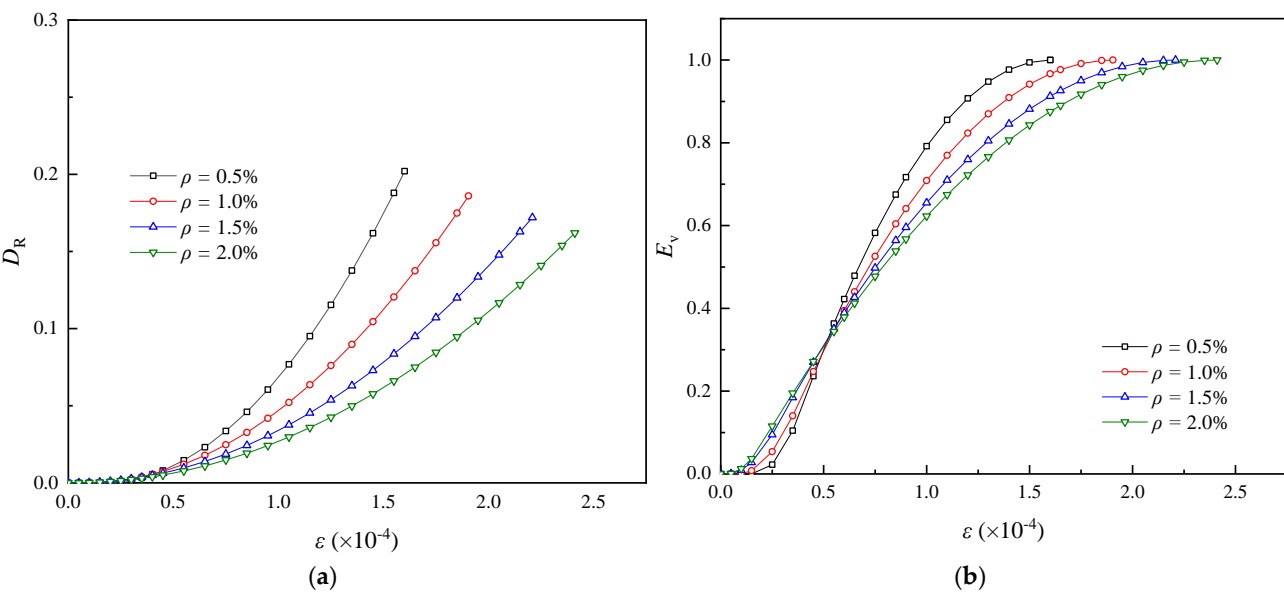

**Figure 12.** Influence of fiber volume fraction on the characteristic parameters: (**a**) $\rho$-$\varepsilon_a$, $\varepsilon_h$, $\varepsilon_b$ curves; (**b**) $H$-$\rho$ curve; (**c**) $E_0$-$\rho$ curve.

**Figure 13.** Evolution curves of damage variable and evolution factor: (**a**) $D_R$-$\varepsilon$ curves; (**b**) $E_v$-$\varepsilon$ curves.

Figure 14 shows the change curves of $S_p$ and $S_{cr}$ with $\rho$. With the increase in the fiber content $\rho$ from 0.5% to 2.5%, $S_p$ increases from 0.161 to 0.345 kPa, $S_{cr}$ increases from 0.266 to 0.548 kPa; and the difference between $S_{cr}$ and $S_p$ increases from 0.105 to 0.203 kPa. The results show that the potential mechanical properties of materials could be fully considered if the energy absorption capacity corresponding to the critical state is adopted.

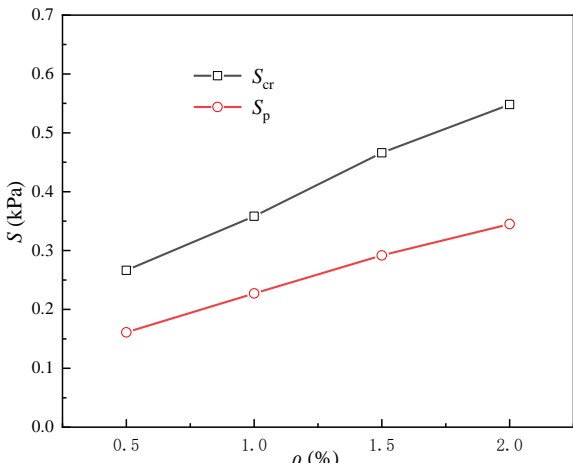

**Figure 14.** Influence of fiber volume fraction on the energy absorption capacity.

The nonlinear stress–strain behavior of fiber concrete is determined by many factors, which include water/cement contents, fiber type and content, aggregate source, additive type, specimen type, loading method and so on. Therefore, although the aforementioned two sets of tests are both for steel fiber concrete specimens, there are great differences both in macroscopic stress–strain behavior and in mesoscopic damage evolution. The experimental results and theoretical analysis show that, in the same group of tests, when only the content of steel fiber is changed, the macroscopic constitutive behavior and mesoscopic damage evolution process of fiber concrete show good regularity with the change of fiber content.

## 5. Conclusions

1. The macroscopic stress–strain behavior (including hardening and softening curves) of concrete under uniaxial tension is a continuous process with deformation and damage evolution. For the traditional segmented constitutive models, two independent expressions are used to describe the pre-peak ascending phase and the post-peak descending phase (taking the peak nominal stress state as the boundary), respectively. Therefore, the link of the mesoscopic damage evolution between the two stages has been isolated artificially. This paper discusses the mesoscopic damage evolution mechanism reflected by the IPBS in detail. The fracture and yield damage modes on meso-scale are considered, and the peak nominal stress state and critical state are distinguished. The uniaxial tensile process is divided into uniform damage phase and local failure phase by the critical state. The uniform damage phase, including the pre-peak ascending segment and a portion of the post-peak descending segment, is the main stage for deformation and damage accumulation and reflects the process from quantitative change to qualitative change. The yield damage mode reflects the development of potential mechanical properties of materials and plays a key role during the whole deformation-to-failure process. Due to the size effect on the local failure phase, the critical state is regarded as the ultimate failure point in the suggested constitutive model.

2. A statistical damage model of fiber concrete under uniaxial tension is established, which considers the fiber enhancement effect. In essence, the addition of fiber changes the composition of the microstructure, restricts the initiation and expansion of microcracks, and also changes the evolution and accumulation process of two damage

modes on a meso-scale. This model contains two kinds of feature parameters ($E$ and $\varepsilon_a$, $\varepsilon_h$, $\varepsilon_b$, $H$) with clear physical meanings, and has the ability to effectively reflect the above changes on meso-scale. Calculations were conducted to simulate the two sets of steel fiber concrete tensile tests in the literature. The experimental and theoretical analysis results show that, when only the fiber content is changed, the shape of the macroscopic nominal stress–strain curve will show a good law of similarity. With the increase of the fiber content, the values of stress and strain corresponding to the peak nominal stress state and the critical state linearly increase, and the curvature of the connecting part of the ascending and descending branch of the nominal stess–strain curve has the changing trend of gradual and orderly. Meanwhile, the characteristic parameters $\varepsilon_a$, $\varepsilon_h$, $\varepsilon_b$, $H$, representing the two types of damage evolution of yield and fracture on a meso-scale, have obvious linear variation law with the change of fiber content. Through this model, the link among the physical mechanism, the mesoscopic damage mechanism and the macroscopic nonlinear constitutive behavior are effectively established.

3.  The macroscopic constitutive behavior of FRC is a complex process of multiple factors. The influence factors include water/cement contents, source of aggregate, fiber type and content, type of additive, specimen size, loading mode, etc. Due to the limitation of the length of articles and test data, only two groups of steel fiber concrete test data are adopted in the validation analysis. Whether this constitutive model could be applicable to the analysis of the influence of other factors on the macroscopic mechanical behavior of fiber concrete, remains to be further researched later.

**Author Contributions:** Methodology, W.B.; data curation, validation and writing—original draft preparation, J.G. and S.H.; writing—review and editing, C.Y. and X.L.; funding acquisition, C.X. All authors have read and agreed to the published version of the manuscript.

**Funding:** This research was funded by National Natural Science Foundation of China, grant number "51679092, 51779095"; National Key R&D Program of China, grant number "2018YFC0406803" and the Science Technology Innovation Talents in Universities of Henan Province, China, grant number "20HASTIT013".

**Institutional Review Board Statement:** Not applicable.

**Informed Consent Statement:** Not applicable.

**Data Availability Statement:** The data presented in this study are contained in this article.

**Acknowledgments:** The author thanks Yao Xianhua and Li Lilie of the School of Civil Engineering and Communication of North China University of Water Resources and Electric Power for their technical support.

**Conflicts of Interest:** The authors declare no conflict of interest.

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
