# Peer review of "Stress–Strain Behavior of FRC in Uniaxial Tension Based on Mesoscopic Damage Model"

_crystals, doi:10.3390/cryst11060689_

Round 1

Reviewer 1 Report

The authors present a work on stress-strain behavior of fiber reinforced concrete in uniaxial tension. In general, it is possible to consider that this work is interesting and current. It is a work based on the statistical treatment of results from other authors, which, however, is well organized and presents interesting conclusions. In my opinion it only needs a few revisions before it is considered for publication.

Here are some comments:

With the reading of the abstract, it is clear what was done in this work. However, the abstract should contain the following 5 basic information: should mention scope, motivation, methodology, results and expected impact of the research.

Figure 1 needs to be formatted in its scale / dimension. In figure 1 (a) the subtitles cannot be read. The acronyms used in the figures must be properly described.

Figures (and tables) should only appear after they are referred to in the text. In the case of figure 1, the figure first appears and only later is it referred to in the text.

The written part needs some revision. For example, the phrase “Since the failure of concrete material essentially attributes to the nucleation and growth of…” needs a grammatical review.

Figures in general (2, 3, 6, etc.) need to be formatted.

Author Response

Authors’ Reply to Reviewers’ Comments

Title (Original): Stress-strain behavior of FRC in uniaxial tension based on mesoscopic damage model

Title (Modified): Stress-strain Behavior of FRC in Uniaxial Tension based on Mesoscopic Damage Model

Authors (Original): Wei-feng BAI, Xiao-feng LU, Jun-feng GUAN *, Shuang HUANG *, Chen-yang YUAN, and Cun-dong XU

Authors (Modified):Weifeng Bai, Xiaofeng Lu, Junfeng Guan *, Shuang Huang *, Chenyang Yuan and Cundong Xu

The authors are very thankful to the editor and reviewers for providing us with careful review and constructive comments. We have revised the paper based on the comments. All changes were highlighted in blue in the revision. Below are our point-by-point responses to the reviewers’ comments and suggestions.

Response to Reviewer 1 Comments

Point 1: The authors present a work on stress-strain behavior of fiber reinforced concrete in uniaxial tension. In general, it is possible to consider that this work is interesting and current. It is a work based on the statistical treatment of results from other authors, which, however, is well organized and presents interesting conclusions. In my opinion it only needs a few revisions before it is considered for publication.

Response 1: Thank you very much for your valuable comments and highly affirmation to the author's research work. We will attach great importance to the relevant opinions and modified carefully.

Point 2: With the reading of the abstract, it is clear what was done in this work. However, the abstract should contain the following 5 basic information: should mention scope, motivation, methodology, results and expected impact of the research.

Response 2: We have supplemented and improved the content of the abstract, focusing on the supplementary of the scope, motivation and expected impact. The modifications have been marked in blue in the paper.

Point 3: Figure 1 needs to be formatted in its scale / dimension. In figure 1 (a) the subtitles cannot be read. The acronyms used in the figures must be properly described.

Response 3: The journal website reformatted the original manuscript, resulting in changes in the scale/format, etc., of some figures in the revised manuscript. We have reworked Figure 1 with improved subtitles and acronyms, as shown in lines 124-125.In addition, we have checked all the figures to ensure the correctness. For example, in Figures 2 to 4, see lines 194-195, 220-231, 268-269, etc.

Point 4: Figures (and tables) should only appear after they are referred to in the text. In the case of figure 1, the figure first appears and only later is it referred to in the text.

Response 4: We found similar problems in 5 places in the paper, which are Figure 1, Figure 5, Figure 7, Table 3 and Figure 11. We have repositioned them so that they lag behind where they first appeared in the article. See lines 124-125, 346-347, 453-463, 506-507, 573-574.

Point 5: The written part needs some revision. For example, the phrase “Since the failure of concrete material essentially attributes to the nucleation and growth of…” needs a grammatical review.

Response 5: We have reorganized the sentences to make the above phrase grammatical, see lines 111-112. In addition, we have proofread the full text and modified the grammar of some sentences. See lines 50, 198, 638-640, 647.

Point 6: Figures in general (2, 3, 6, etc.) need to be formatted.

Response 6: We proofread all the figures and texts in the paper, change all the "Fig n" into "Figure n", and change all the "Eqs" into "Equations". Examples are lines 50, 116, 169-170, and 254-255, etc.

Thanks again to the editors and reviewers for their valuable suggestions. If any deficiencies are found again during the review, please let us know in time and we will make amendments as soon as possible.

Reviewer 2 Report

The paper provides the study of the stress-strain behavior of fiber reinforced concrete (FRC) in uniaxial tension.

The study was carried out on the basis of statistical damage theory and experimental phenomena and considering two kinds of mesoscopic damage mechanisms fracture and yield.

The validity of the proposed model was verified by two sets of test data of steel FRC.

A revision before publishing is recommended according with the following points.

  • - General comment

The authors should improve the literature review citing other research papers presented in international journals dealing with similar topic and extend the introduction with recent and relevant papers. See for example:

  • Mechanical properties and constitutive model of carbon fiber reinforced coral concrete under uniaxial compression. Construction and Building Materials, 263, 120649.

  • On the forced vibration test by vibrodyne. In COMPDYN 2015 conference proceedings(pp. 25-27).

- Minor points

  • Regarding all tables and figures, please change all “Fig. n°” in captions with “Figure n°”. For instance, “Fig 1. Two-stage feature of macroscopic mechanical behavior of concrete under uniaxial tension” will become “Figure 1. Two-stage feature of macroscopic mechanical behavior of concrete under uniaxial tension”

  • Again, regarding figures and tables, please be sure to center all figures and tables as well as all the equations employed

  • Concerning Figure 3 and Figure 6, please reformat the images which seem to be out of all proportion and they are not legible

  • According to the journal guidelines the manuscript should contain: Introduction, Materials and Methods, Results, Discussion, Conclusions (optional). The sections in your work do not seem to meet this requirement. Please try to reorganize your paragraphs in order to also have a Materials and Methods section as well as Results and Discussion sections

  • Please check the conformity of the document with the guidelines of the journal (authors’ names or references for example)

Author Response

Authors’ Reply to Reviewers’ Comments

Title (Original): Stress-strain behavior of FRC in uniaxial tension based on mesoscopic damage model

Title (Modified): Stress-strain Behavior of FRC in Uniaxial Tension based on Mesoscopic Damage Model

Authors (Original): Wei-feng BAI, Xiao-feng LU, Jun-feng GUAN *, Shuang HUANG *, Chen-yang YUAN, and Cun-dong XU

Authors (Modified):Weifeng Bai, Xiaofeng Lu, Junfeng Guan *, Shuang Huang *, Chenyang Yuan and Cundong Xu

The authors are very thankful to the editor and reviewers for providing us with careful review and constructive comments. We have revised the paper based on the comments. All changes were highlighted in blue in the revision. Below are our point-by-point responses to the reviewers’ comments and suggestions.

Response to Reviewer 2 Comments

Point 1: The paper provides the study of the stress-strain behavior of fiber reinforced concrete (FRC) in uniaxial tension. The study was carried out on the basis of statistical damage theory and experimental phenomena and considering two kinds of mesoscopic damage mechanisms fracture and yield. The validity of the proposed model was verified by two sets of test data of steel FRC.

A revision before publishing is recommended according with the following points.

Response 1: Thank you very much for your valuable comments and highly affirmation to the author's research work. We will attach great importance to the relevant opinions and modified carefully.

Point 2: The authors should improve the literature review citing other research papers presented in international journals dealing with similar topic and extend the introduction with recent and relevant papers. See for example:

Mechanical properties and constitutive model of carbon fiber reinforced coral concrete under uniaxial compression. Construction and Building Materials, 263, 120649.

On the forced vibration test by vibrodyne. In COMPDYN 2015 conference proceedings(pp. 25-27).

Response 2: We have added four new literatures on fiber reinforced concrete testing published in the last two years in the journals of "Construction and Building Materials" and "Crystal", and supplemented the related introduction, see lines 41-44, 60.

The numbers of the four documents correspond to 3, 7, 8 and 9 respectively.

As follows:

  1. Zainal, S.M.; Iqbal, S.; Hejazi Farzad; Aziz Farah N. A. Abd.; Jaafar Mohd Saleh. Constitutive Modeling of New Synthetic Hybrid Fibers Reinforced Concrete from Experimental Testing in Uniaxial Compression and Tension. Crystals. 2020, 10(10), 885.
  2. Liu, B.; Zhou, J.; Wen, X.; Hu, X.; Deng, Z. Mechanical properties and constitutive model of carbon fiber reinforced coral concrete under uniaxial compression. Construction and Building Materials. 2020, 263.
  3. Shi, X.J.; Park, P.; Rew, Y.; Huang, K.J.; Sim, C. Constitutive behaviors of steel fiber reinforced concrete under uniaxial compression and tension. Construction and Building Materials. 2020, 233.
  4. Ding, X.X.; Li, C.Y.; Zhao, M.L.; Li, J.; Geng, H.B.; Lian, L. Tensile behavior of self-compacting steel fiber reinforced concrete evaluated by different test methods. Crystals. 2021, 11(3), 251.

Point 3: Regarding all tables and figures, please change all “Fig. n°” in captions with “Figure n°”. For instance, “Fig 1. Two-stage feature of macroscopic mechanical behavior of concrete under uniaxial tension” will become “Figure 1. Two-stage feature of macroscopic mechanical behavior of concrete under uniaxial tension”

Response 3: We proofread all the figures and words in the paper, change all the "Fig n" into "Figure n", and change all the "Eqs" into "Equations". Examples are lines 50, 116, 124-125, 169-170, and 254-255, etc.

Point 4: Again, regarding figures and tables, please be sure to center all figures and tables as well as all the equations employed

Response 4: We checked all the figures, tables and equations to make sure they were all centered.

Point 5: Concerning Figure 3 and Figure 6, please reformat the images which seem to be out of all proportion and they are not legible

Response 5: The journal website reformatted the original manuscript, resulting in changes in the scale/format, etc., of some figures in the revised manuscript. We have reworked Figure 3 and Figure 6 with improved subtitles and acronyms, as shown in lines 230-231,410-411. In addition, we have checked all the figures to ensure the correctness. For example, in Figure 1, Figure 2 and Figure 4, see lines 124-125, 194-195, 268-269, etc.

Point 6: According to the journal guidelines the manuscript should contain: Introduction, Materials and Methods, Results, Discussion, Conclusions (optional). The sections in your work do not seem to meet this requirement. Please try to reorganize your paragraphs in order to also have a Materials and Methods section as well as Results and Discussion sections

Response 6: We have reorganized the articles according to the Journal Guide. The modified manuscript includes: 1. Introduction, 3. Materials and Methods (corresponding to "3. Basis of statistical damage theory" and "4. Statistical damage model for fiber reinforced concrete in uniaxial tension" in the original manuscript), 4. Results and Discussion (corresponding to "5. Empirical verification and damage mechanism analysis" in the original manuscript), and 5. Conclusion. The corresponding structure and level of the titles are also modified. For example, lines 164-165, 181, 205, etc.

Point 7: Please check the conformity of the document with the guidelines of the journal

(authors’ names or references for example)

Response 7: According to the Journal Guide and the format of published articles in the journal, we modify the format of the title of this manuscript, see line 2; and change the format of the author's name, see line 4. The format of references has also been modified, see lines 700-800.

Thanks again to the editors and reviewers for their valuable suggestions. If any deficiencies are found again during the review, please let us know in time and we will make amendments as soon as possible.
